# I-MovE. An intervention to promote movement at childcare centers: Benefits for motor cognitive and socio-emotional development

**Elena Florit[1]☯\*, Tamara Bastianello[1,2]☯\*, Beatrice Andalò[1], Marinella Majorano[1]**

**1** University of Verona Department of Human Sciences, Verona, Italy, **2** University of Padova Department of Developmental Psychology and Socialisation, Padua, Italy

☯ These authors contributed equally to this work.
\* elenaflorit@univr.it (EF); tamarabastianello@unipd.it (TB)

**Data Availability Statement:** All relevant data are within the manuscript and its Supporting Information files.

## Abstract

The present contribution aimed to analyze the effects of a motor program intervention (i.e., I-MovE intervention) implemented indoors and outdoors at nursery school, on children's motor, socio-emotional, and cognitive skills. The study uses a non-randomized pre-post test design. Participants were children attending twenty nursery schools in the North of Italy. The intervention activities were adapted to age: Level 1 activities were addressed to children between 6 and 12 months, and Level 2 activities were addressed to children between 13 and 43 months. Within each level, one group of children developed the intervention indoors (IN-group; Level 1: n = 10; Level 2: n = 104) and another group developed the intervention outdoors (OUT-Group; Level 1: n = 12; Level 2: n = 66). Finally, one additional group was involved as the control group (CONT-Group; Level 1: n = 15; Level 2: n = 98). Children's motor, cognitive, and socio-emotional skills were assessed before and after the intervention by nursery school teachers. The main results showed that the motor intervention promoted children's motor skills development in both groups (i.e., groups implementing Levels 1 and 2 activities) and the cognitive and socio-emotional skills in the older group (i.e., group implementing Level 2 activities), especially the group that performed the intervention outdoors.

## Introduction

Adverse changes of the COVID-19 outbreak further amplified the socio-economic and technological changes which, over the most recent decades, led children to spend less time in physically active activities [1,2] compared to sedentary activities habits (e.g., increased screen time), and less time outside, in green environments [3–5]. These adverse changes have resulted in children's lower motor competence and poorer cognitive, and socio-emotional skills, negative impacts on children's general well-being, social cohesion, and inclusion, as well as public health [6]. The long-lasting movement restrictions in children's lives led to lower motor performances compared with before the lockdown for both girls and boys [5]. Moreover, reduced,

**Funding:** This research was funded by Joint Research 2021 grants promoted by the University of Verona. The funders had no role in study design, data collection and analysis, decision to publish, or preparation of the manuscript.

**Competing interests:** The authors have declared that no competing interests exist.

and restricted regular engagement with natural environments, along with a high level of parental psychological distress, and the temporary interruption of peer relationships during confinement made children more undisciplined and hyperactive, worsening their ability to self-regulate at both cognitive and emotional levels [3].

Given these challenges, the role of motor/physical activities and outdoor education has been reconsidered and promoted to support children's development in multiple domains and their health [e.g., 7–9]. Recent theories are, indeed, supporting interactions across developmental domains, exploring the possibility of resulting in cascading changes throughout periods of developmental transition [10,11]. These theories are the *Ecological* and the *Dynamic Systems* theories [12,13], and the *Embodied Cognition* approach to development [14,15]. In line with these theories, the stimulation of motor skills through the implementation of intervention programs can improve children's functioning in multiple domains due to the interdependent nature of early development [7,16].

The present study aims to describe the results of a structured motor intervention program, the I-MovE intervention, conducted either outdoors or indoors in Italian childcare centers. The program was developed to support nursery school children's motor development and, as a cascade, cognitive and socio-emotional development.

## Motor skills and relations with cognitive and socio-emotional skills

Motor skills are usually classified as being either gross motor skills or fine motor skills [7,17]. Gross motor skills involve coordinating muscle groups or the whole body to obtain balance, moving the trunk and limbs efficiently through space (e.g., walking, running), and moving objects through space (e.g., throwing, catching a ball). Fine motor skills, on the other hand, involve manipulation and visuomotor integration for precisely coordinating small muscle movements (e.g., building with blocks, and tracing). The present contribution focused on motor proficiency related to the development and performance of gross motor skills, especially locomotor skills. In addition, the focus is on fundamental movement skills [18], which support the development of more advanced, specialized movement skills [6,19].

Research on children's motor skills and relations with cognitive and socio-emotional development has focused on constructs such as executive functions (see [20] for a review), self-regulation [16], and socio-emotional competence [21]. Differences in terminology are driven by discipline-specific terminology, yet constructs identified by the different terms partly overlap [20]. In this contribution, we will consider both cognitive and emotional processes (hereafter, attention and early executive functions and socio-emotional competence) with a specific reference to children between 6 and 43 months of age. Attention and early executive functions include processes in which children need to (a) pay attention and switch the focus of attention when needed (cognitive flexibility), (b) remember instructions (working memory), and (c) demonstrate self-regulation (inhibitory control) [22,23]. Socio-emotional skills allow children to self-regulate or control their feelings, thoughts, and behaviors to adapt successfully to family, school, and broader social contexts [24].

The relationships between motor skills and cognitive development have been documented by fundamental theories of child development, such as *Piaget's theory of cognitive development*. More recent studies supported a relationship between motor skills and Executive Functions (EF) in both clinical or at-risk populations [25,26] and populations with typical development [e.g., 7,17,27–29]. Kaiser and colleagues [25] conducted a systematic review of literature showing that school-aged children and adolescents with attention-deficit/hyperactivity disorder (ADHD), especially those with predominantly inattentive disorders, have poorer gross and fine motor skills than their typically developing peers. A more recent systematic review on 3-

to 12-year-olds reported a bunch of studies on preschoolers showing the positive relationship between global motor skills (i.e., measures of both fine and gross motor skills) and EFs (i.e., measures of working memory, inhibition, and shifting [27; see also 30]). Han and colleagues [31] tested and proved the association between early fundamental movement skills (e.g., loco-motor skills) and executive functions (i.e., measures of working memory, inhibition, and cog-nitive flexibility) in 4-year-old children. The correlational studies on 3- to 6-year-olds from urban and rural low-income settings also showed that both inhibition and working memory, but not shifting, were associated with gross motor skills (e.g., locomotor skills) [26]. Veldman and colleagues [17] supported, in a large sample of Australian toddlers, the existence of rela-tions between gross motor skills (e.g., locomotor skills) and cognitive development (sensori-motor development, exploration and manipulation, object relatedness, concept formation, memory, and problem solving), assessed using a global score from a standardized assessment battery. Finally, Gottwald and colleagues [27] provided evidence that motor control and EF (i.e., simple inhibition and working memory tasks) were related in 18-month-olds (n = 53). According to the authors, motor control, and inhibition and working memory are related early in life since movement requires anticipatory control and, therefore, movement and exec-utive function are linked. In sum, the reviewed studies were mainly correlational and most of them investigated the relationship between gross motor and cognitive skills in children between 3 and 6 years of age. However, some of these studies provide evidence supporting the existence of associations between motor skills and EF (i.e., attention, working memory, and inhibition) in both nursery children and toddlers [27,28]. These associations can be explained by *Embodied cognition theories* and *Dynamic System Theory* that support the idea that motor experiences in the first years of life provide children with more opportunities to explore the environment which, in turn, enhances their cognitive abilities [13,28].

Motor skills have also been studied in relation to socio-emotional skills in both clinical pop-ulations [21,32] and populations with typical development [e.g., 30]. For example, research analyzing the relationship between children's motor skills and socio-emotional skills focused on children with movement problems defined as Developmental Coordination Disorders (DCD; e.g., [21]). DCD is a neuro-developmental disorder characterized by diminished fine and/or gross motor coordination, which co-occurs with socio-emotional difficulties [33]. Although DCD is usually diagnosed in mid-to-late childhood, available evidence has investi-gated the relationship between individual differences in motor coordination and socio-emo-tional competence in preschoolers to establish when motor coordination problems emerge. King-Dowling and colleagues [21] compared a group of 3-to-6-year-olds children with dimin-ished motor coordination with a group of typically developing peers matched for age and intel-ligence, using a behavioral parent-reported checklist to identify emotional-behavioral problems. They found that preschool children with diminished motor coordination showed both higher internalizing problems in the form of withdrawal and externalizing problems, such as aggression, compared with the group of typically developing peers. Similar results were also reported by Piek and colleagues [34] in typically developing 4-year-old children. An exploratory study by Gandotra and colleagues [30] found significant positive associations between gross motor skills and prosocial behavior assessed through a teacher-rated question-naire in 3–5 years old children. More recently, Cavadini and colleagues [35] implemented a large-scale study on 706 French preschool children aged 3 to 6 to assess relations between motor skills (e.g., locomotor skills), socio-emotional competence, and mathematical perfor-mance. Previously trained nursery teachers observed and tested the children's gross motor skills and socio-emotional competence and were continuously monitored via a digital plat-form. Cavadini and colleagues [35] found that gross motor skills influence preschool children's social behavior and their ability to understand emotions. Finally, Cheung and colleagues [36]

showed that fine and gross motor skills contributed to the development of socio-emotional skills in children with (in preschool and kindergarten) and without (in preschool only) disabilities.

Taken together, findings from previous studies support the existence of an association between gross motor and socio-emotional skills in 3-to-6-year-olds. According to Cavadini and colleagues [35], a possible explanation for this association may lie in the fact that locomotor skills are a privileged way of learning how to interact with others and experiencing emotions, especially in school and through social play. Children with low motor abilities, therefore, may have fewer relational opportunities to develop their understanding of emotions and social behavior [37]. All these health outcomes, in turn, have positive effects on school learning and success, as well as social cohesion and inclusion [20,35].

## Motor interventions for promoting motor, cognitive and socio-emotional development

Although children's motor skills emerge because of normative development, motor competence develops more rapidly through instruction [38]. Motor skills develop especially during what Clark and Metcalfe named the Fundamental patterns period, starting from the second year of life, and lasting until about seven years, for most children [19] (p. 15). Motor skills interventions provide children with the instruction and practice necessary to develop motor skills.

To date, motor skill interventions were mainly carried out with 3–5 years old children or older children [for reviews, see 39–43] while published intervention studies in toddlers are lacking [40]. An editorial by Libertus and Haus [44] discussed the role of intervention in improving motor skills in typically and atypically developing infants. In particular, the author reported studies focusing on interventions in infants under three years of age [e.g., 45,46]. Both programs aimed at making infants independent on specific skills (i.e., object engagement and exploration [45] and sitting [46]) and seem to be beneficial for children's motor development. Systematic reviews and meta-analysis of the literature by Van Capelle and colleagues [43] and Wiek et al., [42] analyzed the effects of gross-motor intervention programs on fundamental movement skills (e.g., locomotor skills). Only interventions on children aged 2–6 years and equal to or greater than 4 weeks duration were considered to exclude short-term effects. Findings revealed that most of the studies focused on children aged 3 to 5 years with significant effects (small-to-large) on fundamental movement skills (see also Strooband and colleagues [41] for a systematic review of the interventions on fine motor skills in 0–6-year-olds). Tortella and collegaues [47] developed an intervention for improving gross-motor skills (and to a lesser extent fine motor skills) in 5-year-olds Italian children. The results showed that the experimental group who practiced gross-motor activities in the playground for 1 hour a week for 10 weeks improved significantly in 4 out of the 6 gross motor tasks (and in none of the fine motor tasks). Therefore, within the literature, there is a consensus on the role of motor intervention programs in improving children's gross motor skills. However, no other developmental outcomes were analyzed. In other words, what is missing from the reviewed empirical findings on motor intervention programs is the consideration of potential effects on other developmental domains–i.e., "dynamic connections across domains" [44, p. 3].

To our knowledge, very few studies analyzed whether motor skills interventions delivered in early childhood had benefits on motor, cognitive, and socio-emotional development at the same time. Notable exceptions are the recent study of Hudson and colleagues [7] and Mulvey and colleagues [29]. Hudson and colleagues [7] tested whether participating in a comprehensive motor skills curriculum (16 motor skills sessions over 8 weeks) was related to

improvements in motor skills and Executive Functions (and early numeracy skills) of 53, 3–5 years old children. The intervention was developed in small groups of 4 children to teach and practice gross (e.g., locomotor) and fine motor skills and involved planning, monitoring, and controlling coordinative actions. The authors found significant small to moderate treatment effects for motor skills and EF (and early numeracy skills) with stronger effects for inhibitory control than working memory. Similarly, Mulvey and colleagues [29] showed that a proven gross motor skill intervention (the Successful Kinesthetic Instruction for Preschoolers; SKIP), conducted twice a week (30 mins per section) for a total of 6 weeks, improves (medium-to-large effects) preschoolers' gross motor skills and EF (i.e., behavioral regulation but with inherent motor demands; for a program on motor skills and executive functions in 8 years old children see also [48]). The limited number of studies analyzing the benefits of motor interventions on multiple developmental domains in children younger than 3 years of age represents an important gap in both theoretical and practical knowledge. Motor skill intervention seems more appropriate and socially engaging to promote children's skills early in development due to the interdependent nature of early development [7,16].

Literature also suggested that structured fine and gross motor skill intervention supports children's motor and cognitive skills while outdoor free play programs (e.g., no formalized or structured instruction or physical education) fail to promote these skills [16,26,49]. Interestingly, the various health benefits of motor intervention appear to be higher during physical education taught outdoors rather than indoors [6]. Arguably, motor interventions carried out outdoors may be more beneficial due to the additional benefits of outdoor environments (see for instance [50]). The exposure to, the time spent, or the subjective experiences (restorativeness) in natural environments alone can both restore and enhance the mind and behavior leading to socio-emotional and cognitive benefits [51–53 for reviews]. For instance, Carrus and colleagues [54] compared socio-emotional competence and attention performance of small groups of children aged 18–36 months after exposure to outdoor green spaces or indoors built-up spaces. Results consistently showed that exposure to nature in educational settings promotes psychological restoration, strengthens children's cognitive and emotional resources, and increases the quality of children's social interaction. Similar results were reported for groups of toddlers and preschoolers that attended childcare centers where a continuous Outdoor Education (OE) program was carried out versus childcare centers implementing traditional educational activities [55–57]. The results of these studies can be explained by the *attention restoration theory* [58] according to which the natural environment could help in supporting skills in the cognitive domain; interacting with the natural environment helps restore attentional energies and supports in pursuit of results. In addition, the experience in nature could also reduce tiredness and fatigue enabling children to activate good cognitive regulation [59]. The *stress reduction theory* [60] argues that exposure to a green environment would help in reducing stress levels at both physiological (e.g., decrease in heart rate) and cognitive levels (e.g., higher concentration). In other words, natural environments would allow individuals to replace negative emotions with positive ones, facilitating recovery from stressful events.

## Aims and hypotheses

According to existing evidence, the positive effects of motor activity on child development are well-known, but very few intervention studies have confirmed these benefits in children in their first three years of life. The latter is a crucial period for laying the foundations of individual and social growth [57] (p. 869), thus this topic deserves special attention. Particularly, previous literature suggested that being involved in structured motor intervention activities may

bring benefits, especially for the development of the motor and cognitive domain (e.g., gross motor abilities, locomotor skills; [7,49]). Finally, it is well-established that outdoor educational activities bring benefits to children's development [6,9]. Based on this empirical foundation, structured motor activities conducted outdoors, in green space, in a group of nursery children should be much more beneficial for the development of cognitive and socio-emotional competencies.

Since all these effects and patterns of influence have not been explored in children in the first three years of life, the present study aimed to assess the effects of the I-MovE intervention, a structured intervention program based on gross motor activities, which was implemented indoors (IN-group) or outdoors (OUT-group) in infants and toddlers attending nursery schools in the North of Italy. A third group of children of the same age as the experimental IN- and OUT-groups, was included as a business-as-usual control group (i.e., CONT-group).

More specifically we aimed to analyze:

1) the effects/impact of the I-MovE intervention on children's a) motor, b) cognitive, and c) socio-emotional development. First, we expected growth for all children, in their motor, cognitive, and socio-emotional skills. Due to a maturation effect, such growth was expected both for children in the experimental group (i.e., those children attending I-MovE) and in the control group (i.e., those children who did not attend the intervention). However, based on previous evidence (see for example, 7,47) a statistically significant interaction effect between Time (Pre vs Post) and Group (Experimental vs Control) was expected in the motor and cognitive skills of children attending the intervention. Moreover, based on correlational evidence [32], we also hypothesized a significant effect of the intervention on socio-emotional skills.

2) The effects of the motor-based intervention in the group of children who implemented the motor activities indoors vs the group who implemented the same motor activities outdoors. In other words, we explored if growth in children's motor (cognitive, and socio-emotional) development was explained by the structured motor activities per se or by the fact that the activities were also implemented indoors vs outdoors. We expected benefits for children's motor, cognitive, and socio-emotional skills, especially if the program was implemented outdoors [e.g., 9]. In detail, we expected better performances in the cognitive-related areas [58] and socio-emotional-related areas [60], for children implementing the program outdoors.

## Materials and methods

### Participants

A target group of 14 childcare centers located in several provinces in Northern Italy was involved to take part in the intervention for a total of about 257 children with typical development (teachers and parent report). Among those children, 192 children ($M_{age}$ = 23.9, $SD$ = 7.84) who attended at least 10 sessions (i.e., 50% of the sessions of the intervention program; see "Fidelity of the implementation" section for more details) were retained as part of the experimental group. Within the experimental group, two sub-groups were identified: one who experienced the proposed motor program intervention indoors (i.e., IN, $n$ = 114; $M_{age}$ = 24.2, $SD$ = 8.03), and one who experienced it outdoors (i.e., OUT group, $n$ = 78, $M_{age}$ = 23.5, $SD$ = 7.60).

Additional six childcare centers with 113 children of around 22.2 months ($SD$ = 8.12), were involved as the control group (CONTROL group). This group adopted a traditional curriculum in line with national guidelines for nursery schools including educational practices and growing experiences to promote children's general growth, creativity, and autonomy [61,62].

Children of the three groups (i.e., experimental IN, experimental OUT, and control) were recruited from about mid-February 2022 (after the final study's approval by the ethics

committee of the host institution) until mid-March 2022. The children were involved in the study only after obtaining informed consent from their families.

Children in each of the three groups (i.e., experimental IN, experimental OUT, and control), we further divided into two age levels: Level 1 = children between 6 and 12 months ($M$ = 10.5, $SD$ = 1.92) and Level 2 = children older than 13 months (13–43 months; $M$ = 25.1, $SD$ = 6.78). This additional distinction was necessary because the activities of the intervention were adapted to infants younger than 12 months in the experimental (IN and OUT) and children older than 13 months (see Intervention Program section). The number of children in Levels 1 and 2 of the experimental IN, experimental OUT, and control groups is reported in Table 1.

The childcare centers involved to recruit the control and experimental groups were in the same provinces and paired by characteristics as to the Socio-economic Status (SES) of catchment areas served by the childcare centers, the number of children attending each childcare center, availability of outdoor spaces, and quality of the school in terms of structure and processes.

The selected sample is a convenience sample since children were not randomly assigned to the intervention or control groups because of the difficulties of some nursery schools in adhering to and implementing the intervention. To overcome this issue, we controlled some relevant intervening variables. To compare children from different childcare centers, we asked the coordinators to fill in the SVANI scale [63] (see Instruments sections for more details on scales and questionnaires) to assess the quality of the involved educational structures. In addition, nursery teachers were administered a questionnaire on the frequency of physical and outdoor activities carried out at the childcare centers. Parents were asked to fill out a demographic questionnaire and were administered the Restorativeness scale [51] to assess the individual perception of restorative qualities (fascination, being away, coherence, scope) of the outdoor/green spaces of the childcare centers. Table 1 reports the data for each of the three groups (IN, OUT, and CONTROL) for Level 1 and Level 2 and comparisons among the three groups at each level.

Based on these data, we can exclude differences among structures involved in recruiting experimental (IN or OUT) and control groups in terms of the quality of schools [$F(2,15)$ = .583, $p$ = .570, $\eta^2_p$ = .07], and demographic characteristics of the participants (e.g., child's age and parent's level of education; see Table 1). However, we cannot exclude differences in the frequency of motor and outdoor activities and restorativeness score and in hours of attending childcare (for Level 2 only). For this reason, these variables were considered as covariates in the main analyses. The study was approved by the ethics committee of the host institution (ethical approval code: 2022_03).

## Design and procedure

The study was carried out from mid-February 2022 until the end of July 2022 and consisted of a non-randomized pre-test (T0 phase, before starting the experimentation), an intervention phase, and a final post-test (T1 phase, 2 months later). Fig 1 graphically represents the study design and instruments used.

All children were tested using the same tools before (T0) and after (T1) the intervention (see Fig 1). The instruments selected for being used at pre-test and post-test were mainly rating scales and parent/teacher reports that require a relatively short time of administration. Instruments were adapted to be completed by the nursery teachers via a digital platform (Google Forms, children's codes were used for privacy protection issues) and with the supervision of the academic team due to the persistence of the Covid-19 pandemic and difficulties entering

**Table 1. Characteristics of participants in Level 1 (children younger than 12 months) and Level 2 (children older than 13 months).**

| Level 1 | | | | | | |
|---|---|---|---|---|---|---|
| | IN | OUT | CONTROL | $F$ | $p$ | *partial $\eta^2$* | Post-hoc comparisons |
| N | 10 | 12 | 15 | | | | |
| *Children's charatceristics* | | | | | | | |
| Age | 9.5 (2.12) | 11.30 (1.60) | 10.7 (1.84) | 2.51 | .097 | .13 | - |
| Gender | | | | | | | |
| Female (%) | 30% | 50% | 40% | $\chi^2 = 0.929, p = .628$ | | | - |
| Age of entry to childcare | 7.89 (1.69) | 10.2 (3.05) | 9.36 (5.09) | 1.13 | .342 | .09 | - |
| Hours attending childcare | 5.96 (1.54) | 7.49 (1.24) | 7.64 (1.80) | 3.33 | .054 | .22 | - |
| *Parent's characteristics* | | | | | | | |
| Parimary caregiver's age | 34.1 (4.20) | 40 (5.40) | 35.3 (6.58) | 3.18 | .060 | .22 | - |
| Parimary caregiver's educational level (a) | 3.11 (1.45) | 2.60 (0.84) | 2.57 (1.40) | 0.530 | .596 | .04 | - |
| *Motor and outdoor* | | | | | | | |
| Motor activities frequency(b) | 3.50 (0.71) | 3.75 (0.45) | 2.70 (0.45) | 8.49 | .001 | .35 | CONT vs IN* CONT vs OUT** |
| Outdoor activities frequency (b) | 2.50 (0.00) | 3.50 (0.00) | 2.53 (0.74) | 15.6 | < .001 | .49 | CONT vs OUT*** IN vs OUT*** |
| Restorativeness | 73.3 (24.0) | 85.3 (15.2) | 77.3 (17.90) | 0.945 | .403 | .08 | - |

| Level 2 | | | | | | |
|---|---|---|---|---|---|---|
| | IN | OUT | CONTROL | $F$ | $p$ | *partial $\eta^2$* | Post-hoc comparisons |
| N | 104 | 66 | 98 | | | | |
| *Children's charatceristics* | | | | | | | |
| Age | 25.6 (6.86) | 25.8 (5.90) | 24 (7.18) | 1.85 | .159 | .01 | - |
| Gender | | | | | | | |
| Female (%) | 45.54% | 61.19% | 48.27% | $\chi^2 = 4.04, p = .133$ | | | - |
| Age of entry to childcare | 13.11 (4.88) | 12.86 (5.51) | 12.88 (5.56) | .053 | .948 | < .01 | - |
| Hours attending childcare | 5.91 (1.89) | 6.80 (1.48) | 7.38 (1.53) | 12.51 | < .001 | .11 | CONT vs OUT*** IN vs OUT ** |
| *Parent's characteristics* | | | | | | | |
| Parimary caregiver's age | 36.18 (5.91) | 36.48 (5.11) | 35.79 (4.87) | 0.340 | .712 | < .01 | - |
| Parimary caregiver's educational level (a) | 3.14 (1.23) | 3.34 (1.31) | 3.35 (1.91) | 0.357 | .700 | < .01 | - |
| *Motor and outdoor* | | | | | | | |
| Motor activities frequency(b) | 3.13 (0.64) | 2.86 (0.77) | 2.96 (0.71) | 2.951 | .054 | .02 | IN vs OUT |
| Outdoor activities frequency (b) | 3.201 (0.42) | 2.468 (0.32) | 2.97 (0.75) | 39.87 | < .001 | .24 | CONT vs IN *** CONT vs OUT * IN vs OUT *** |
| Restorativeness | 82.32 (15.03) | 73.79 (15.44) | 84.63 (12.01) | 12.62 | < .001 | .21 | CONT vs IN*** IN vs OUT ** |

Note: For each level, the number of children, their characteristics (i.e., age, gender, age of entry to childcare, hours attending childcare), their parent's characteristics (i.e, primary caregiver's age and education level), motor and outdoor practices and restorativeness scores are reported.

a) years of education treated as a categorical variable, 1 = middle school; 2 = high school; 3 = University degree; 4 = Post Lauream.

b) Frequency treated as categorical variable, 1 = never; 2 = 2–3 times/week; 3 = once per day; 4 = more times per day.

* In post-hoc comparisons, p-values were corrected with Bonferroni; p < 0.5

**p < .01

***p < .001.

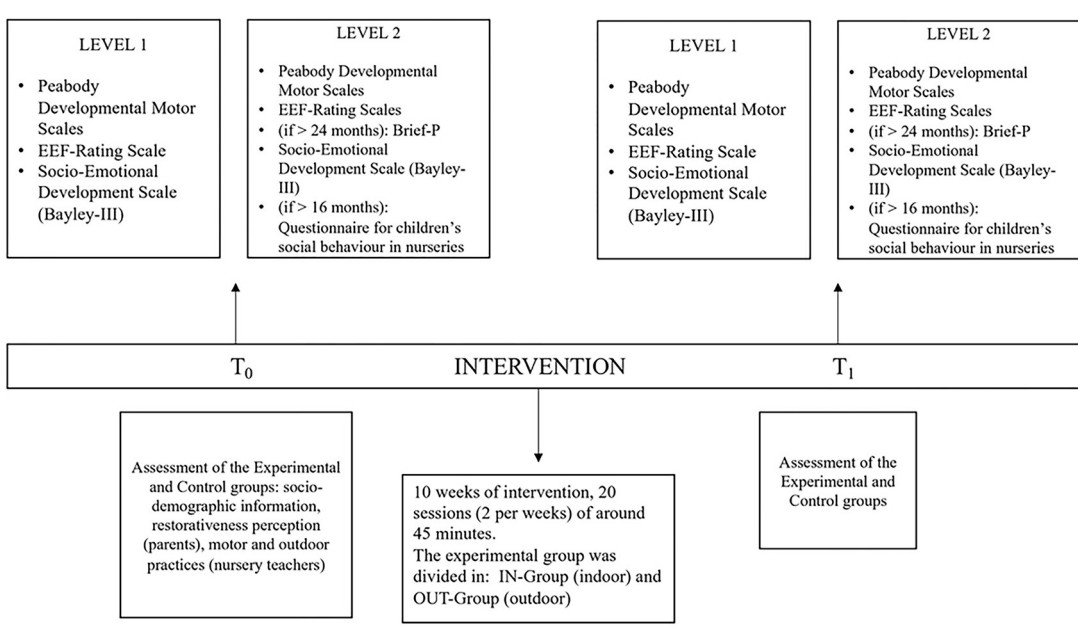

**Fig 1. Graphical representation of the study design and instruments.**

nursery schools. Nursery teachers of both the experimental and control groups were trained and monitored by the academic team to administer the motor, cognitive, and socio-emotional assessments at the pre-test and post-test stages. Codes were used to fill in the questionnaires and the scales, therefore all data are anonymized.

The next sessions describe the intervention, its implementation with fidelity checks, the instruments used, and the measures collected at T0 and T1. The intervention was devised by the academic team which included experts in developmental and educational psychology, pedagogical and in motor sciences, and implemented by nursery teachers. At the pre-test phase, all nursery teachers of the target childcare centers received two intensive training sessions (for a total of 14 hours over two days) aimed at improving their knowledge of children's motor development and its importance for promoting children's overall development and well-being. During these meetings, nursery teachers were also trained and sensitize on how to use appropriate behaviors and on how to conduct the sessions of the intervention. Nursery teachers were also provided with video recordings accompanied by detailed written instructions on how to administer the instruments Online meetings/phone calls on demand were provided during the implementation of the intervention.

## Motor intervention program

The I-MovE intervention program was built to promote the development of gross-motor skills, specifically, fundamental movement skills in 0–3 years old children. It has two specific aims. Firstly, it wants to promote the ability of the children to approach specific competencies and try to learn how they work (the exploration phase). Secondly, the child "trains" to learn how to manage specific competencies and internalize their functioning mechanisms (the development phase). The program included 20 sessions that took place twice a week over ten weeks. Each session focuses on a specific competence with particular attention to the development of fundamental movement skills. In detail:

1. Weeks 1 and 6 focused on activities promoting the exploration and development of postural control—i.e., the positions that the body can assume;

2. week 8 focused on activities promoting the exploration and development of the movement of the limbs- i.e., the movements of a body segment such as bending the lower limbs;

3. weeks 2, 3, 4, 5, 7, 9, and 10 are focused on activities promoting the exploration and development of fundamental movement skills—i.e., the first expressions of movement of the human being from conception, including walking, running, jumping, crawling, climbing, rolling, throwing and catching.

Activities were presented as play-based and divided into basic and advanced, based on their level of complexity. Basic activities were addressed to children up to 12 months of age (hereafter Level 1 group) and advanced activities were addressed to children aged more than 13 months (hereafter, Level 2 group). Some of these activities were common to both levels.

Each session was organized into three parts: 1) an initial warm-up at the end of which nursery teachers shared and agreed with children on the rules of the activities (4–6 minutes for both groups); 2) a central part in which children practiced activities to explore and develop gross-motor skills, specifically, fundamental movement skills (15–20 minutes for the Level 1 group; 20–25 minutes for the Level 2 group); 3) a final moment to relax (4–6 minutes for both groups). Therefore, the entire session was expected to last, on average, 30–35 minutes for Level 1, and 35–40 minutes for Level 2. The duration of the whole intervention program is greater than the duration recommended to exclude short-term effects and it is in line with the average duration of previously conducted motor programs (see, for example, Hudson and colleagues [6], Tortella et al. [47] for empirical evidence; see Riethmuller and colleagues [40], Wick et al. [42] and Van Capelle et al. [43] for reviews and a meta-analysis).

Table 2 describes one of the activities (program details and the full set of activities are available upon request to the first author), "The rolling slide", delivered by nursery teachers during the second week of the intervention and addressed to both groups. The activity was aimed at promoting the ability to roll as a fundamental movement skill.

**Fidelity of the implementation.** The fidelity of the implementation of the intervention was checked by using a diary created ad hoc for this research project and observations of videotaped lessons.

During the intervention, the diary was compiled by the nursery teachers. It was organized into two main parts. In the first part, nursery teachers reported for each weekly session: the date, the duration of the session, and if it was carried out indoors or outdoors. In addition, for

**Table 2. Example of the activity "The rolling slide" (second week of intervention).**

| Target motor skill | Development of the ability to roll around the longitudinal axis in passive form. |
|---|---|
| Setting | Two adults must be present.<br>A mat and a resistant blanket/sheet must be placed in the play area |
| Description of the activity | Each adult is placed on one of the two sides of the mat and holds two corners of the blanket/sheet which leans on the mat.<br>Taking turns, the children lie down on the blanket/sheet (supine or prone, as they like).<br>One of the two educators lifts the corners to raise the blanket/sheet and create a slope, on which the child will roll.<br>When the child will be moved to the opposite side of the blanket/sheet, the other adult will lift the corners to raise the blanket/sheet and create a slope from that side. |
| Equipment | • One high and soft mat<br>• Blanket/sheet |
| Age group | Levels 1 and 2 (from 8 months) |

each activity, there were several questions relating to the difficulties encountered during the preparation and implementation of the proposed activities (e.g., "Was the space for the activity/ game easily set up? YES NO; If not, why?"). The main data collected for the first part showed that the mean duration of each session was of 40.4 minutes with no variations based on the child's age (children in Level 1 performed sessions of around 39 minutes, SD = 11.3; children in Level 2 of around 41 minutes; SD = 8.69). This confirms that nursery teachers respected the instructions provided in the guideline of the activities. The more qualitative analysis of encountered difficulties showed that most of the concerns on the implementation of the activities regard the respect of the turns to allow each child to participate and the respect of the guidelines of the activities. Moreover, difficulties were reported more often during the activities carried out during the first 5 weeks of intervention than the activities carried out during the final 5 weeks. This may be explained by the acquisition of some experience in organizing and managing activities by the nursery teachers. In the second part, nursery teachers reported on the presence/absence of each participant in each session of the intervention. The analysis showed that 257 children in total took part in the intervention. On average, these children were involved in 13.8 (SD = 5.07) sessions (69%) over 20 sessions of the intervention. Indeed, part of the children did not attend all the intervention sessions due to Covid 19 restrictions. Sometimes, childcare centers were also forced to close due to the Covid outbreak. Based on these data, we excluded children who did not attend at least 50% of the program, which are 68 from our starting sample of 257 children. This choice was made to assure children of a reasonable level of participation.

Six childcare centers (representing 43% of the total structures) were also asked to videotape a session during the central part (i.e., week 6) of the intervention, and a reliability analysis was conducted. An independent coder analyzed each video recording by considering three macro-categories regarding the implementation of the activity (i.e., adherence to the guidelines provided to nursery teachers and respect of the time limit); sensitivity of the teacher (i.e., nursery teachers showing appropriate behaviors, for example when children show difficulties, based on the training received), and children participation (percentage of involved/excluded children or presence of children who did not agree to participate in the activity) from which a reliability index was calculated for each childcare center. The average score is 77.71%. This score indicates that the activities carried out by the nursery teachers in week six affordably meet the standard criteria. Thus, the implementation of the intervention can be considered affordable.

### Instruments

**Socio-demographic information, restorativeness of green areas outside the nursery school, frequency of outdoor and motor activities carried out at the childcare centers and the SVANI scale.** Two questionnaires were administered to parents. The first one was the socio-demographic questionnaire providing information on children's age, nursery school attendance, primary caregiver's age, and education. The second questionnaire is the Perceived Restorativeness Scale [64] which assessed the parent's subjective experience of the green spaces (i.e., garden) outside the school. In detail, this scale aims to assess the degree to which the natural environment can be instrumental in the restoration of mental, emotional, and physical well-being [52]. Parents were asked to assess how much 11 statements apply to their experience of the nursery school garden. On a 0 to 10-point Likert scale, where 0 = not at all, 6 = rather much, and 10 = completely. The higher the score, is higher the restorativeness perception (i.e., the feeling of restoring mental, emotional, and physical well-being) for that place. For the present study, we considered the total score as a sum of each item's score.

Nursery teachers were asked to fill out a questionnaire to evaluate the frequency of the motor and outdoor activities with children in nursery school (1 = never; 2 = 2–3 times/week;

3 = once per day; 4 = more times per day). They were asked to do so by referring to the autumn/winter and spring/summer seasons. The final score of motor and outdoor activities was the average frequency of motor and outdoor activities during the two periods, respectively. The data for these measures are reported in the Participants section and Table 1.

Finally, each childcare coordinator was asked to complete the SVANI scale. The SVANI is a 7-scales observational tool, made up of 37 items to assess the school's quality in terms of furnishings and materials, personal care and routine, the language used, learning activities, interactions, organization of activities, and adult needs.

**Motor Development—Peabody Developmental Motor Scales (all children).** The locomotor scale of the Peabody Development Motor Scales (Second Edition) (PDMS-2, [65]; Italian adaptation by [66]) evaluates gross motor skills (such as jumping, rolling, climbing, and descending stairs). The locomotor scale is composed of 91 items and presents behaviors typical of children between 5 months and 42 months. Based on the age of the child under observation, the teacher referred to the corresponding item and assigned a score from 0 to 2 based on the child's motor skills, according to the following indications: the child received a score of 2 when he/she performed the task according to the criteria required by the item; score 1 was assigned when the child's behavior approached the criteria of the item but does not fully reach them; the score 0 was assigned when there was no evidence that the skills specified by the item were emerging. During the assessment, the nursery teachers had the opportunity to consult the "PDMS-2—Administration Guide", in which each item is described in detail and is associated with the criteria necessary for assigning the score. The items proposed to recreate specific situations in which the teacher must verify the child's behavior and assess whether it corresponds to one of the descriptions given in the items. A specific observational situation is therefore created for each motor behavior required by the questionnaire. For this tool, raw total scores were considered. The internal consistency of the locomotor scale of the normative sample is excellent ($\alpha$ = .96) [66].

**Cognitive Development—EEF-Rating Scales (all children).** The EEF-Rating Scales assessed children's early executive functions (EEF, in the domains of attention, self-regulation, and cognitive flexibility) and have recently been validated on a wider group of 656 children between 6 and 36 months [67]. The EEF- Rating Scales are presented in various forms according to the child's age. The first form evaluates a child's attention and EEF in the first year of life (0–12 months) and it is comprised of 10 items assessing basic attentional skills (n = 7 items, i.e., the ability to selectively deal with objects, events, or interests), self-regulation (n = 1 item), and cognitive flexibility (n = 2 items). The second form measures a child's attention and EEF in the second year of life (13–24 months). It is made up of 10 items that evaluate basic attentional skills (n = 5 items), self-regulation (n = 3 items), and cognitive flexibility (n = 2 items). Finally, the third form measures a child's attention in the third year of life (25–36 months), It is made up of 10 items for the assessment of attentional skills (n = 3 items), the assessment of self-regulation (n = 4 items), and the examination of cognitive flexibility (n = 3 items). The frequency with which behavior occurred was evaluated by the nursery teacher on a 0- to 5-point Likert scale where 1 = never, 2 = at least once in the last month, 3 = at least two times in the last month, 4 = at least once a week and 5 = every day. From the first analysis for its validation, it emerged that the three scales of the EEF have a good internal consistency (on average $\alpha$ = .81).

**Cognitive Development–Behavior Rating Inventory of Executive Functions-Preschool Version (for children > 24 months).** The Brief-P (Behavior Rating Inventory of Executive Function—Preschool Version [68]; Italian adaptation by [69]) is a screening tool for possible executive dysfunction in preschool age. As a standardized rating scale, the Brief-P is designed to measure behavioral manifestations of executive functions in children from two to five years of age, within the context of the everyday school environment. The Brief-P is divided into 5

independent clinical scales that measure different aspects of executive functions and represent the 5 domains mainly involved in executive dysfunctions detected in preschool age: inhibition, shift, regulation of emotions, working memory, and planning /organization. For this project, only three scales have been selected: inhibition, shift, and working memory, for a total of 43 items. The following question was initially asked the nursery teacher: "During the last 6 months, how often has each of the following behaviors been a problem?"; subsequently, various behavioral manifestations of executive functions that could represent a problem are listed, enclosed in the 43 items. The nursery teachers evaluated the frequency with which each behavior represents a problem, choosing among the options often (S), sometimes (Q), or never (M). The scores obtained represent the degree of difficulty the child has: the higher the score, the higher the level of difficulty the child exhibits. The internal consistency of the selected scales filled out by the nursery teachers of the normative sample is excellent (inhibition α = .94; shift α = .90; and working memory α = .94).

**Socio-Emotional Development—Bayley-III Socio-Emotional Development Scale Caregiver Report (all children).** Bayley III (Italian version of the Bayley Scales of Infant and Toddler Development—Third Edition, [70]) is an individual evaluation scale to identify 0–3 years old children with developmental delay. Specifically, the scales investigate five developmental domains: cognitive, language, motor, socio-emotional and adaptive behavior. For this project we administered only the scale referred to the socio-emotional domain, which evaluates, through a questionnaire, the child's emotional state, measuring what he/she usually does and what he/she might be able to do in relationships with adults. In particular, the socio-emotional scale is an age-related scale designed for children from birth to 42 months of age. It is composed of 39 items, referring to the main socio-emotional milestones such as the ability to comprehend emotional signals, the use of emotional expressions, and the elaboration of a range of feelings using words and other symbols. Nursery teachers rated the frequency of behavior on a 0- to 3-point Likert scale, where 0 stands for unable and 3 stands for always. The internal consistency of the socio-emotional scale is good (α = .84).

**Socio-Emotional Development–Questionnaire for children's social behaviour in nurseries (for children > 16 months).** The Questionnaire for children's social behavior in nurseries [71,72] was used to assess the social and relational competencies of children older than 16 months. It consists of 22 questions that evaluated the frequency with which a certain social behavior at school occurs based on a 0- to 3-point Likert scale where 0 = rarely; and 3 = always. The questionnaire covers three main domains of social competence: 1) emotional competence (EC, 10 items, items 4, 7, 12, 13, 14, 15, 16, 19, and 20); 2) social engagement (SE, 8 items, 1, 2, 3, 8, 9, and 22); 3) and aggressiveness (AG, 4 items, 5, 6, 10, and 11). The appropriate items (items 1, 3, 6, 8, 9, 22) were reversed. Following Lanciano et al. [73], three items were excluded because they did not load on relevant factors (items: 17, 18, and 21). Higher scores indicate higher social skills concerning peers in nursery school. The internal consistency of the three sub-scales is, on average, good (α = .80).

## Data analysis

Preliminarily, the sample distribution was checked for all the considered dependent variables (DV). To test the effects of the I-MovE intervention program on experimental groups vs. control group (aim 1) and its effect in the group implementing the motor activities indoors vs the group implementing the activities outdoors (aim 2), two separate sets of analyses were run (first, for Level 1 and then, for Level 2) on the motor, cognitive, and socio-emotional skills. For both Level 1 and Level 2, we conducted a series of Linear Mixed-Effects Models (LMMs). Mixed models were used to take into account the nested structure of the data since

observations are nested in individuals (i.e., repeated measures of relevant DVs) who are nested in childcare centers and classes. Estimation problems prevented the fit of planned models with both random intercepts and slopes. As suggested by Barr et al. [73], non-converging models were dealt with by progressively simplifying the random effects structure until convergence was reached, resulting in our case in a random-intercept-only model. Measures of motor, cognitive, and socio-emotional skills were considered as continuous dependent variables, Group (with three levels, experimental IN, experimental OUT, and CONTROL) and Time (with two levels, T0, and T1) as fixed effects, Subjects (ID) and schools' classes as a random effect, and frequency of motor and outdoor activities, the restorativeness index of nursery schools and hours of attending childcare (only for Level 2) as covariates.

We run seven LMMs (DVs were the scores of the Peabody Developmental Motor Scales, EEF-rating Scales, Brief-P, Socio-Emotional Development Scale—Bayley III, and Questionnaire for children's social behavior in nurseries) for children of Level 2 and three LMMs (DVs were the scores of the Peabody Developmental Motor Scales, EEF-rating Scales, and Socio-Emotional Development Scale—Bayley III) for children of Level 1. The percentage of variance explained by each model was expressed by marginal R2 and conditional R2, which refer, respectively, to the variance explained by the fixed effects and the global variance of random and fixed effects. If a significant interaction term was found, a simple slope analysis was explored by considering the parameter estimates (to address both aims 1 and 2). Descriptive statistics were reported for each DV for both groups at T0 and T1. For each model, simple group effects at T0 were run to exclude the presence of starting significant differences. Jamovi Version 2.3.22 [74,75] was used to perform the analyses.

To test whether the sample sizes of the two groups (Level 1 and Level 2) in the present study play a role in limiting the significance of each Linear Mixed-Effects Model, a series of post-hoc power analyses were conducted. These analyses were performed using G*power [76], with an $\alpha = 0.05$, an effect size (f) derived from the $\eta^2_p$ of the interaction term Group*Time of each model, and the exact number of participants for Level 1 and Level 2.

## Results

First, Skewness tests were run and revealed no abnormal distributions (range Level 1: 0.279–0.281; range Level 2: -0.017–1.29). Descriptive statistics were run for each group at both times, and they were reported in Table 3. For all DV we excluded the presence of significant differences among groups at T0 (all $ps > .05$).

### Effects of the I-MovE intervention program and its modality of implementation (indoor vs. outdoor) on children's motor skills

**Peabody Developmental Motor Scales—Level 1.** The marginal $R^2$ and conditional $R^2$ of the model were .30 and .89 respectively. The ANOVA test showed a significant main effect of Time [$F(1;30.69) = 54.47$, $p < .001$, $\eta^2_p = .47$], and interaction between Group and Time [$F(2;30.7) = 3.64$, $p = .038$, $\eta^2_p = .18$; observed power $1 - \beta = .99$]. The main effect of Group was not significant ($p = .517$, $\eta^2_p = .06$). Fig 2 showed performances on the Peabody motor scale of the three groups at pre- (T0) and post-test (T1).

To explain the interaction term, we look at fixed effects of the parameters estimates that showed that both the group that carried out the intervention outdoors [$b = -13.882$, $t(18.35) = -2.865$, $p = .010$] and indoors [$b = -19.599$, $t(18.35) = -3.854$, $p = .010$] increased their motor skills from T0 to T1 if compared to the control group. The experimental groups' slopes did not significantly differ from each other ($p = .159$). Therefore, results confirmed that the intervention had a significant, similar effect on both the motor skills of the OUT and the IN groups.

**Table 3. Mean and Standard Deviations of the scores obtained by children in Level 1 and 2 in the motor, cognitive, and socio-emotional assessment instruments at T0 and T1.**

| | T0 | | | T1 | | |
|---|---|---|---|---|---|---|
| | **IN** | **OUT** | **CONTR** | **IN** | **OUT** | **CONTR** |
| | **Motor skills** *Peabody Developmental Motor Scales* | | | | | |
| Level 1 | 26.30 (7.96) | 59.17 (18.9) | 46.47 (17.61) | 51.60 (9.00) | 78.09 (17.30) | 60.33 (22.93) |
| Level 2 | 79.52 (14.83) | 88.58 (18.67) | 87.30 (19.67) | 96.20 (21.31) | 105.00 (19.69) | 94.51 (20.88) |
| | **Cognitive skills** *EEF-Rating Scales* | | | | | |
| Level 1 | 22.70 (3.129) | 26.33 (6.75) | 24.73 (5.861) | 23.40 (4.22) | 25.83 (4.09) | 24.73 (6.02) |
| Level 2 | 25.12 (4.92) | 24.23 (5.44) | 24.91 (5.45) | 24.46 (7.27) | 25.61 (5.31) | 24.53 (5.30) |
| | *BRIEF-P* | | | | | |
| Level 2—Inhibition | 25.77 (7.70) | 23.43 (7.53) | 27.22 (8.99) | 23.65 (7.45) | 22.25 (6.61 | 26.30 (8.20) |
| Level 2—Shifting | 13.94 (3.58) | 12.55 (2.89) | 14.39 (3.86) | 13.81 (3.14) | 12.55 (3.24) | 15.70 (4.41) |
| Level 2—Working Memory | 25.69 (6.38) | 22.91 (6.42) | 26.37 (8.23) | 23.99 (5.72) | 22.16 (6.81) | 26.20 (8.09) |
| | **Socio-emotional skills** *Socio-Emotional Development Scale (Bayley-III)* | | | | | |
| Level 1 | 61.11 (9.72) | 98.00 (33.70) | 70.73 (14.83) | 76.00 (12.28) | 116.00 (28.23) | 77.60 (21.40) |
| Level 2 | 110.89 (30.20) | 116.53 (29.75) | 94.71 (40.28) | 125.29 (30.27) | 140.00 (27.36) | 99.39 (41.74) |
| | *Questionnaire for children's social behaviour in nurseries* | | | | | |
| Level 2 | 32.90 (7.80) | 34.84 (8.39) | 32.92 (8.52) | 38.68 (8.73) | 40.73 (6.60) | 36.22 (8.11) |

This indicates that either the group that carried out the intervention outdoors or indoors significantly increased their motor skills from T0 to T1.

**Peabody Developmental Motor Scales—Level 2.** The marginal $R^2$ and conditional $R^2$ of the model were .11 and .89, respectively. The ANOVA test showed a significant main effect of Time [$F(1;137.94) = 127.21$, $p < .001$, $\eta^2_p = .23$], (note that here and in the following analyses of the variance, the df were approximated for Satterthwaite), and a significant interaction between Group and Time [$F(2;127.70) = 6.153$, $p = .003$, $\eta^2_p = .05$, observed power $1 - \beta > .99$]. The main effect of Group ($p = .439$, $\eta^2_p = .01$) was not significant. Covariates did not affect the DV in any way (all $ps > .05$). Fig 3 showed performances on the Peabody motor scale of the three groups at pre- (T0) and post-test (T1).

To explain the interaction term, we look at fixed effects of the parameters estimates that showed that either the group that carried out the intervention outdoors [$b = 7.017$, $t(125-76) = 3.248$, $p = .001$] and indoors [$b = 6.040$, $t(129.75) = 2.733$, $p = .007$] increased their motor skills from T0 to T1 if compared to the control group. The experimental groups' slopes did not significantly differ from each other ($p = .660$). This indicates that either the group that carried

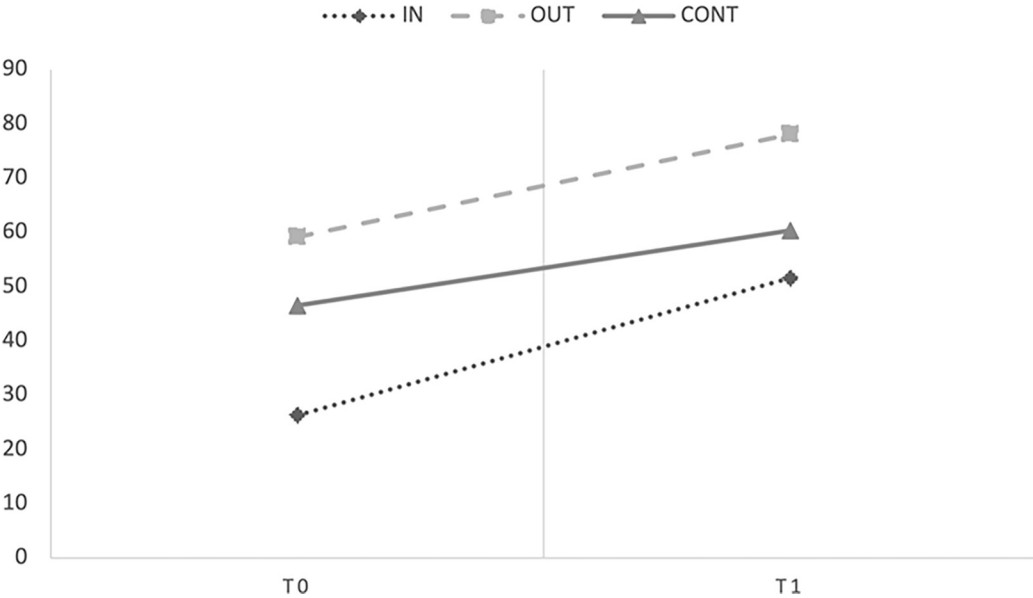

**Fig 2. Mean scores obtained by Level 1 children in the Peabody motor scale at T0 and T1.**

out the intervention outdoors or indoors increased their motor skills from T0 to T1 significantly and to a similar extent.

## Effects of the I-MovE intervention program and its modality of implementation (indoor vs. outdoor) on children's cognitive skills

**EEF-Rating Scales–Level 1.** The marginal $R^2$ and conditional $R^2$ of the model were .08 and .56, respectively. The ANOVA test showed no significant main effects of Time ($p = .811$,

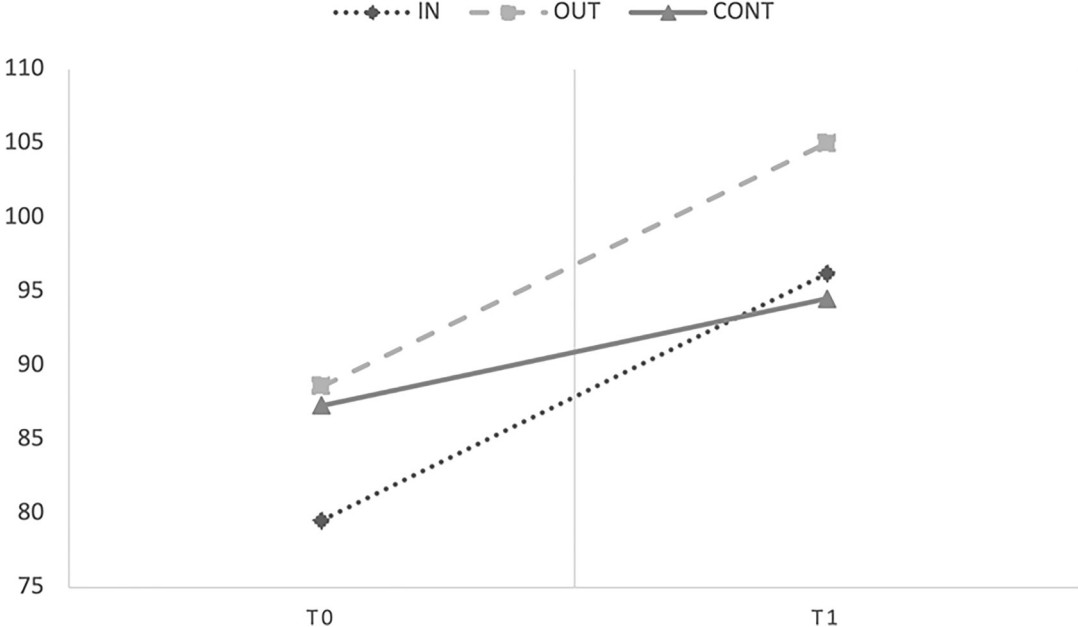

**Fig 3. Mean scores obtained by Level 2 children in the Peabody Developmental Motor Scale at T0 and T1.**

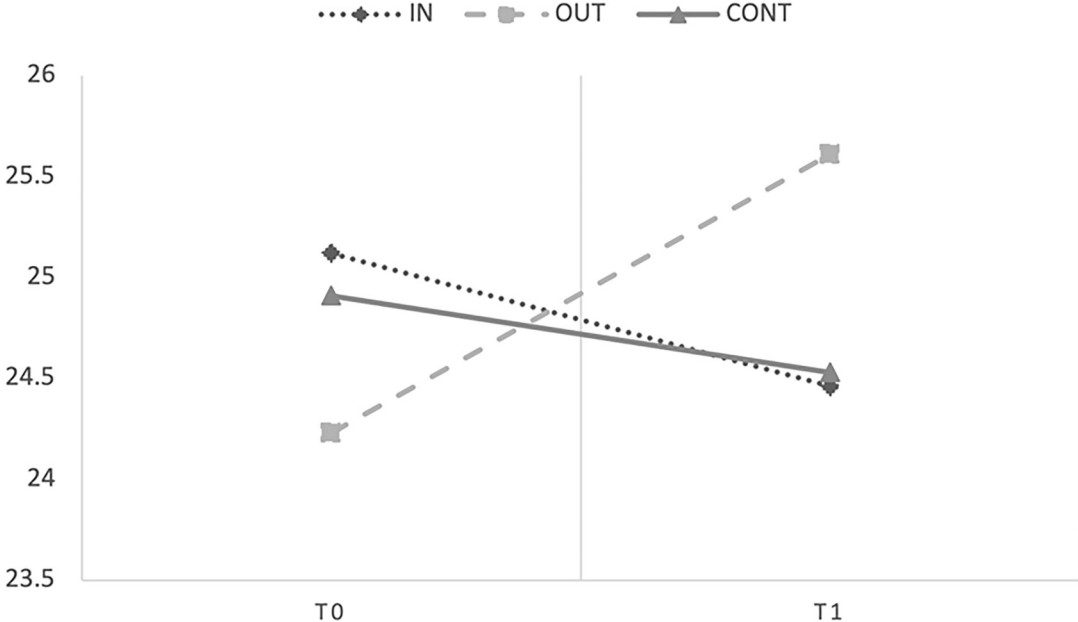

**Fig 4. Mean scores obtained by Level 2 children in the EEF rating scales at T0 and T1.**

$\eta^2_p$ = .02), group ($p$ = .717, $\eta^2_p$ = .01), and interaction between Time and Group ($p$ = .388, $\eta^2_p$ = .02, observed power 1 $-\beta$ = .30) (see Table 3 for descriptive statistics). Covariates did not affect the DV in any way (all ps > .05).

**EEF-Rating Scales–Level 2.** The marginal $R^2$ and conditional $R^2$ of the model were .04 and .49, respectively. The ANOVA test showed no significant main effects of Time [$F$ (1;197.29) = 0.012, $p$ = .911, $\eta^2_p$ = .01] and Group [$F$(2;41.25) = 0.149, $p$ = .862, $\eta^2_p$ = .01], and a significant interaction between Group and Time [$F$(2;197.38) = 5.535, $p$ = .005, $\eta^2_p$ = .01, observed power 1 $-\beta$ > .99]. Covariates did not affect the DV in any way (all *ps* > .05). Fig 4 showed performances on the EEF scales of the three groups at pre- (T0) and post-test (T1).

To explain the interaction term, we look at fixed effects of the parameters estimates that showed that only the group that carried out the intervention outdoors ($b$ = 2.46, $t$(197.46) = 2.226, $p$ = .027) increased their EEF from T0 to T1 if compared to the control group. The scores of the group that carried out the intervention indoors did not significantly differ from those of the control group ($p$ = .295). The experimental groups' slopes significantly differ from each other [$b$ = 3.52, $t$(196.65) = 3.299, $p$ = .001]. Indeed, the group that implemented the intervention outdoors, but not the group implementing the activities indoors, improved their EEF skills. These results confirmed that the intervention had a significant and positive effect on the EEF skills of the OUT group but not on the IN group.

**Brief-P–Level 2.** The results of the Brief-P are available for children who at T0 were older than 24 months ($n$ = 154).

*Inhibition*. The marginal $R^2$ and conditional $R^2$ of the model were .06 and .76, respectively. The ANOVA test showed a significant main effect of Time [$F$(1;124.60) = 7.33, $p$ = .008, $\eta^2_p$ = .05], but neither a significant main effect of Group ($p$ = .183, $\eta^2_p$ = .01) nor a significant interaction between Time and Group ($p$ = .090, $\eta^2_p$ = .02, observed power 1 $-\beta$ > .99). Covariates did not affect the DV in any way (all *ps* > .05). Estimated marginal means for Time showed that there was a general decrease in the inhibition scores from T0 ($M$ = 25.38, $SE$ = 0.756) to T1($M$ = 24.00, $SE$ = 0.736). Therefore, regardless of group membership, children's inhibition skills measured as behavioral manifestations improved from T0 to T1.

*Shifting*. The marginal $R^2$ and conditional $R^2$ of the model were .14 and .66, respectively. The ANOVA test showed a significant main effect of Group [$F(2;24.65) = 6.195$, $p = .007$, $\eta^2_p = .01$], but neither a significant main effect Time ($p = .071$, $\eta^2_p = .01$) nor a significant interaction between Time and Group ($p = .133$, $\eta^2_p = .04$, observed power $1 - \beta > .99$). With the only exception of the restorativeness scale [$F(1;129.92) = 5.185$, $p = .023$], covariates did not affect the DV in any way (all $ps > .05$). Estimated marginal means for Group showed that there was a difference among the performances of the three groups in the shifting scale of the Brief-P. In particular, the OUT group obtained lower scores for shifting ($M = 11.98$, $SE = 0.793$) compared to the IN group ($M = 13.59$, $SE = 0.628$) and the control group ($M = 15.72$, $SE = 0.737$). Therefore, the shifting skills of children in the OUT group were better compared to the other two groups.

*Working Memory*. The marginal $R^2$ and conditional $R^2$ of the model were .08 and .76, respectively. The ANOVA test showed that the main effects of Time ($p = .228$, $\eta^2_p = .04$), group ($p = .297$, $\eta^2_p = .01$), and the interaction between Time and Group ($p = .115$, $\eta^2_p = .03$, observed power $1 - \beta > .99$) were not significant (see Table 3 for descriptive statistics). Covariates did not affect the DV in any way (all ps > .05).

## Effects of the I-MovE intervention program and its modality of implementation (indoor vs. outdoor) on children's socio-emotional skills

**Socio-Emotional Development Scale (Bayley-III)–Level 1.** The marginal $R^2$ and conditional $R^2$ of the model were .31 and .91, respectively. The ANOVA test showed a significant main effects of Time [$F(1;19.52) = 15.372$, $p < .001$, $\eta^2_p = .20$], but neither a significant main effect of Group ($p = 290$, $\eta^2_p = .01$) nor a significant interaction between Group and Time ($p = .154$, $\eta^2_p = .06$, observed power $1 - \beta = .75$). As regards the covariates, the motor activity frequency affects the model's results [$F(1;14.15) = 8.19$, $p = .012$]. Estimated marginal means for the Time showed that there was an increase in the socio-emotional skills from T0 ($M = 69$, $SE = 7.19$) to T1($M = 78.95$, $SE = 7.96$). Therefore, overall, children's social skills measured as behavioral manifestations improved from T0 to T1.

**Socio-Emotional Development Scale (Bayley-III)–Level 2.** The marginal $R^2$ and conditional $R^2$ of the model were .20 and .81, respectively. The ANOVA test showed significant main effects of Time [$F(1;192.84) = 67.04$, $p < .001$, $\eta^2_p = .09$] and Group [$F(2;46.32) = 5.447$, $p = .008$, $\eta^2_p = .02$], and the interaction between Group and Time [$F(2;192.92) = 7.674$, $p < .001$, $\eta^2_p = .01$, observed power $1 - \beta > .99$]. Covariates did not affect the DV in any way (all $ps > .05$). Fig 5 showed performances on the Bayley socio-emotional scale of the three groups at pre- (T0) and post-test (T1).

To explain the interaction term, we look at fixed effects of the parameters estimates that showed that the group that carried out the intervention outdoors [$b = 18.27$, t(192.98) = 3.916, p < .001] increased their socio-emotional skills from T0 to T1 compared to the control group. For the indoor group, the intervention had only a marginally significant effect [$b = 8.133$, t(194.52) = 1.89, p = .060]. In addition, the experimental groups' slopes significantly differ from each other [$b = 8.133$, $t(194.52) = 2.267$, $p = .025$]. Therefore, the results confirmed that the intervention had a significant effect on the socio-emotional skills of the OUT group.

**Questionnaire for children's social behavior in nurseries—Level 2.** The results of the questionnaire are available for children who at T0 were older than 16 months ($n = 262$). The marginal $R^2$ and conditional $R^2$ of the model were .16 and .69, respectively. The ANOVA test showed a significant main effect of Time [$F(1;188.94) = 117.43$, $p < .001$, $\eta^2_p = .13$], but neither a significant main effect of Group ($p = .050$, $\eta^2_p = .01$) nor a significant interaction between Group and Time ($p = .181$, $\eta^2_p = .01$, observed power $1 - \beta > .99$). Covariates did not affect the

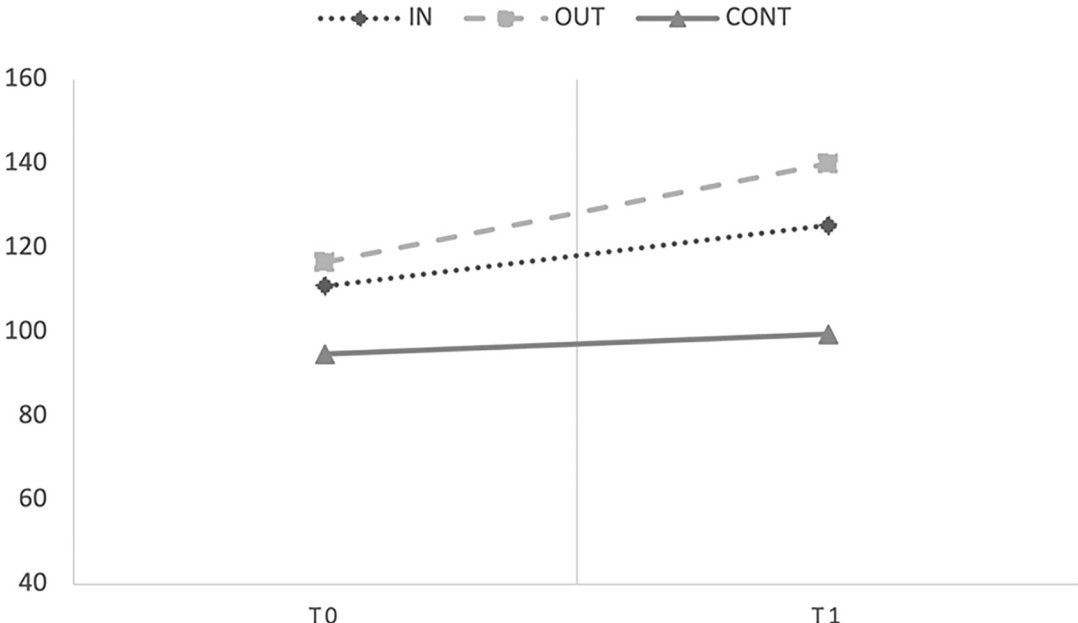

**Fig 5. Mean scores obtained by Level 2 children in the Socio-Emotional Development Scale Caregiver Report of the Bayley III Scales at T0 and T1.**

DV in any way (all $ps > .05$). Estimated marginal means for Time showed that there was a general increase in the questionnaire scores from T0 ($M = 32.86$, $SE = 0.917$) to T1($M = 38.59$, $SE = 0.908$). Therefore, overall, children's social skills measured as behavioral manifestations improved from T0 to T1.

## Discussion

The present study aimed to test the effects of the I-MovE motor skills intervention program on nursery children's motor, cognitive, and socio-emotional skills. The motor intervention was implemented by nursery teachers indoors or outdoors during the spring of 2022, with the persistence of the Covid-19 pandemic. Both experimental groups were compared with the business-as-usual control group. The three groups come from childcare centers that were comparable in educational quality and differences in relevant control variables were also controlled in the analyses. The main results showed that the motor intervention promoted children's motor skills development in the younger and older group (i.e., Levels 1 and 2) and the cognitive and socio-emotional skills in the older group (i.e., Level 2). These results have relevant theoretical, methodological, and practical implications for children's development and well-being. These implications and the limitations of the study will be discussed in the following paragraphs.

### The I-MovE motor intervention program promoted motor skills development in infants and toddlers in indoor and outdoor settings

The program significantly improved gross motor skills in infants younger than 12 months and toddlers older than 13 months implementing the program indoors and outdoors. Thus, these findings support the effectiveness of the I-MovE intervention in promoting gross motor skills in children younger than those considered in existing studies reviewed in the Introduction. Previously published motor interventions focused mainly on children from 3 to 5 years of age

and older [7,29,47; see also 43,44 for review]. These interventions considered locomotor skills and had a similar duration and dosage to the I-MovE intervention (e.g., 30-min sessions twice weekly, for 8 weeks in [7] and 6 weeks in [29]; one hour with 30 minutes of structured activities in the playground, once a week for ten consecutive weeks [47]) and effects of similar magnitude [7,42,43]. Therefore, our results suggest that motor development, at least when fundamental movement skills and locomotor skills are considered, can be promoted effectively even earlier than 3 years of age by an intervention with such characteristics. Furthermore, according to Van Capelle [43] and Wick [42], the duration of the I-MovE intervention allows the exclusion of short-term effects on motor development. Finally, according to our findings, the required knowledge, and practical instruments to implement the program within the typical daily routines of childcare centers, can be provided to nursery teachers to support their work in this specific developmental domain.

Partly in contrast with our expectations, children's motor development benefited similarly from the intervention program both when it was implemented indoors and outdoors. This result adds to previous evidence [16,26,49,55] in suggesting that structured motor activities presented through play modalities may bring benefits to infants' and toddlers' motor development in different settings.

In conclusion, findings of the present study showed that, although children's motor skills emerge due to normative development, motor competence develops more rapidly through instruction [18,38]. Structured motor skills interventions implemented in educational contexts [43] may provide young children with the instruction and practice necessary to develop these motor skills through play modalities.

## The I-MovE motor intervention program promoted cognitive and socio-emotional development in toddlers, especially in the outdoor setting

The results for the older group of children, but not for the younger group, showed, at least partly, that the I-Move motor program intervention had small positive effects on multiple domains of development (i.e., cognitive, and socio-emotional domains). Specifically, concerning the cognitive domain, two were the main findings. First, a significant effect of the intervention on children's attention skills and precursors of executive function skills emerged for the group of children implementing the intervention outdoors. Instead, we failed to find a significant effect on these skills for children who performed the intervention indoors. This result is partly in line with evidence from the few existing studies on toddlers [e.g., 27] and with our expectations and evidence that the various health benefits of motor intervention appear to be higher during physical activity taught outdoors rather than indoors [6]. This result may be interpreted in light of the *Attention Restoration Theory* [58] or the *Stress Reduction Theory* [60], according to which, contact with nature allows people to restore resources consumed in activities and tasks that require voluntary attention and, consequently, to recover from a situation of cognitive fatigue. This, in turn, supports in pursuit of results and enables children to activate good cognitive regulation [59]. In other words, the outdoor environment seems to widen the effects of the intervention per se on attention and early executive functions. Second, this pattern was not confirmed when we considered children's executive functions as assessed by using the Brief-P [57,58]. This can be accounted for by considering some of the differences between the two scales. For instance, the Brief-P, but not the EEF Rating scale, focused on problematic behaviors. More importantly, most of the items of the EEF Rating scale focused on attention skills and to a lesser extent on the precursors of EF. The different results obtained for the two scales can be explained considering findings obtained in adults [53], and the assumption of the *Attention Restoration Theory* [58], according to which the environment

mainly affects tasks that require more attentional resources. Differences across scales and settings might also contribute to explaining why our results are only partly in line with evidence of the effectiveness of motor intervention implemented in preschool center settings on EF skills [7].

We also found that the I-MovE intervention—implemented especially outdoors—proved effective in promoting older children's socio-emotional skills assessed through the Socio-Emotional Questionnaire of the Bayley scales [57]. However, we failed to find a significant effect of the intervention on children's social skills as assessed with the Questionnaire for children's social behavior in nurseries [61,62]. These results are partly in line with correlational evidence that supported a relationship between motor skills and socio-emotional competence in older typically developing preschoolers [35,36] and clinical populations [32].

In addition, our results are also in line with the *Stress Reduction Theory* and studies on the effects of OE [55,57], which suggests that being exposed to natural environments allows the individual to replace negative emotions with positive ones, facilitating recovery from stressful events. We can hypothesize two explanations for the different results found for the Socio-Emotional Questionnaire of the Bayley scales and Questionnaire for children's social behavior in nurseries. First, the Socio-Emotional Questionnaire of the Bayley scales [70] focuses on the relations between the child, the adult, and peers while Questionnaire for children's social behavior in nurseries [71,72] focuses on the relations between the target child and their peers. Thus, it may be that the activities of the intervention which were led by the nursery teacher and did not focus on collaborative games or group plays might mainly benefit social-emotional competencies as assessed by considering relations with adults. Second, it may also be the case that the positive effects of motor interventions on the social-emotional competencies as assessed by considering relations with peers are evident later, in preschool children [e.g., 35,36]. For instance, Cheung et al. [36] found a relation between motor skills such as walking and running and socio-emotional competence with peers in preschool children with typical development. Social interactions with peers require the development of complex socio-emotional skills that toddlers have not yet developed.

The I-MovE intervention did not prove significant effects on the cognitive and socio-emotional domains of the youngest group of children. This may be related to several factors. First, the sample size of children in Level 1 increased the difficulty of identifying the significant effects of the intervention in other domains. In particular, the post-hoc power for the EEF-Rating Scales–Level 1 was low. Second, the duration of the intervention or the number of sessions was not sufficiently long to induce changes in these domains in infants.

To conclude, our findings partly support the cascading effect of the I-MovE motor intervention program on developmental domains other than the motor domain in the older group of children. The sizes of the effects of the interaction terms of the models testing the effect of the I-MovE intervention on the cognitive and socio-emotional domains also tend to be lower than those for the motor domain, although in the range reported in previous interventions [7]. In other words, this study adds to previous evidence on preschoolers [e.g., 7,35] by showing that formalized motor skills intervention may be suitable for promoting cognitive and socio-emotional development and specifying which setting is more effective.

## Limits and conclusions

First, the present study utilized a convenience sample. Although relevant intervening variables were controlled in the analyses, children were not randomly assigned to the intervention or control groups. This was due to the difficulties faced by some childcare centers in the implementation of the intervention and some others' inability to adhere. Second, the proportion of participants younger than 12 months (i.e., children in Level 1) was low. Future studies should

dig deeper into the present findings, especially those concerning the cognitive domain, by involving wider groups of infants. Third, the effects of conducting the program Outdoors (i.e., the combined effect of motor intervention activities and green) should be further confirmed. To understand this, future studies could compare the performances of a group of children conducting the I-MovE motor intervention outdoors with a group of children implementing spontaneous activities in an Outdoor Education (OE) program (e.g., nurseries in the wood). Fourth, the program was implemented during a period in which the Covid-19 pandemic persisted and still affected the typical daily routines in nurseries, researchers' access to schools, children's absence from school, and school/class closure. Overall, this resulted in difficulties in data collection and implementation of the I-MovE motor intervention and, consequently, in data loss. Fifth, these difficulties were paired with difficulties linked to the very limited number of assessments available for infants and toddlers from 0–3 years of age. For this reason, the number of children in the different analyses was not constant but varied. For instance, the Brief-P could be administered to children older than 24 months. Consequently, analyses were conducted on a reduced number of participants (i.e., 154 children). Sixth, the Covid-19 pandemic restrictions forced the adaptation of the procedure to administer the instruments online at T0 and T1. Nursery teachers were supported in the administration through means of preliminary meetings, videos, and online meetings/phone calls, but we cannot completely exclude that the online administration might have raised some issues during the assessment of some children and the use of some scales. More qualitative analyses carried out to check for the fidelity of implementation, revealed that nursery teachers found relatively few difficulties in respecting the guidelines of the activities. Nevertheless, future interventions based on the I-MovE program must address this issue through further adaptation of guidelines. Finally, future research should examine the degree to which treatment effects will persist or whether continued intervention efforts are necessary to maintain effects over time.

Despite these limitations, we believe that our work has relevant theoretical, methodological, and practical implications. At a theoretical level, this study represents an original contribution to the field of developmental and educational psychology, since it is the first to investigate the effects of a motor intervention program on multiple developmental domains (i.e., motor, cognitive, and socio-emotional) in children younger than 3 years of age. This study, therefore, contributes to analyzing the relationship among these three domains, and it investigates the influence of other related factors such as the role of the (green) environment. More importantly, the results of the present study suggest the I-MovE intervention is appropriate for promoting children's skills early in development, due to their interdependent nature [7,16]. At a methodological level, the present study is innovative because it systematically monitored the implementation of the intervention. Specifically, the integrated use of the diary and videotapes of a target session to check the fidelity of the implementation of the intervention supported the scientific rigor of this work and its findings.

At a practical level, the findings of the present study may bring benefits to psychologists, nursery school teachers, parents, and different settings. Evidence of the effects of the I-MovE intervention on various children's skills can significantly contribute to improving psychologists' awareness of the crucial role of early motor training in strengthening different developmental domains. The I-MovE intervention program may constitute an innovative tool addressed to nursery school teachers to increase early motor and, to some extent, cognitive and socio-emotional skills. Therefore, this contribution can enhance nursery teachers' awareness of the possibility of implementing specific motor programs at a very young age, even before children acquire confidence and autonomy in fundamental movement skills. Participation in the project has also provided teachers with new tools to monitor children's early skills: all teachers—under the supervision of experts—had the chance to use tools to assess children's

motor, cognitive, and socio-emotional development. In other words, the project has also contributed to improving teachers' observational skills. Relatedly, their knowledge of children's milestones in motor development and awareness of the benefits of the promotion of motor development increased during the intervention. Furthermore, parents have benefitted from the program indirectly: nursery teachers have reported that they documented the motor activities conducted indoors and outdoors and spread them to parents. After the intervention, childcare centers inform parents about the activities carried out with children through different communicative modalities (e.g., events, group meetings, posters). Furthermore, based on parent's feedback, teachers also reported that some children reported on and proposed the games they played at school in the home environment. Based on the evidence that parental involvement is crucial to ensuring the transfer of knowledge from the setting of the intervention to the home environment [40], this study opens several potentials for promoting motor education practices within the home setting, too. Finally, our findings increased the awareness of psychologists, nursery teachers, and parents on the role of the setting of the intervention, specifically on the exposure to natural environments, in determining different developmental outcomes.

In conclusion, therefore, the I-MovE intervention may contribute to supporting the early development of crucial skills. This, in turn, may have positive impacts on skills that support later school learning and success [35] and, more widely, may contribute to promoting children's general well-being, social cohesion, and inclusion, as well as public health [6].

## Supporting information

**S1 Dataset.**
(XLSX)

## Author Contributions

**Conceptualization:** Elena Florit, Beatrice Andalò, Marinella Majorano.

**Data curation:** Tamara Bastianello.

**Formal analysis:** Elena Florit, Tamara Bastianello.

**Funding acquisition:** Elena Florit, Beatrice Andalò.

**Investigation:** Elena Florit, Tamara Bastianello, Beatrice Andalò.

**Methodology:** Elena Florit, Tamara Bastianello.

**Project administration:** Elena Florit.

**Resources:** Beatrice Andalò.

**Supervision:** Elena Florit, Marinella Majorano.

**Visualization:** Tamara Bastianello.

**Writing – original draft:** Elena Florit, Tamara Bastianello.

**Writing – review & editing:** Elena Florit, Tamara Bastianello, Beatrice Andalò, Marinella Majorano.

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
