## [Decision Letter · Decision Letter 0]

31 Aug 2023

PONE-D-23-11988I-MovE. An intervention to promote movement at childcare centers: benefits for motor cognitive and socio-emotional developmentPLOS ONE

Dear Dr. Bastianello,

Thank you for submitting your manuscript to PLOS ONE. After careful consideration, we feel that it has merit but does not fully meet PLOS ONE’s publication criteria as it currently stands. Therefore, we invite you to submit a revised version of the manuscript that addresses the points raised during the review process.

We look forward to receiving your revised manuscript.

Kind regards,

Josie Booth

Academic Editor

PLOS ONE

Journal Requirements:

"This research was funded by Joint Research 2021 grants promoted by the University of Verona."             

3.In your Data Availability statement, you have not specified where the minimal data set underlying the results described in your manuscript can be found. PLOS defines a study's minimal data set as the underlying data used to reach the conclusions drawn in the manuscript and any additional data required to replicate the reported study findings in their entirety. All PLOS journals require that the minimal data set be made fully available. For more information about our data policy, please see http://journals.plos.org/plosone/s/data-availability.

Additional Editor Comments:

Thank you for your patience with the review process. Please attend to all reviewer comments.

Reviewers' comments:

Reviewer's Responses to Questions

**Comments to the Author**

1. Is the manuscript technically sound, and do the data support the conclusions?

Reviewer #1: Yes

Reviewer #2: Partly

2. Has the statistical analysis been performed appropriately and rigorously? 

Reviewer #1: I Don't Know

Reviewer #2: Yes

3. Have the authors made all data underlying the findings in their manuscript fully available?

Reviewer #1: No

Reviewer #2: No

4. Is the manuscript presented in an intelligible fashion and written in standard English?

Reviewer #1: Yes

Reviewer #2: Yes

5. Review Comments to the Author

Reviewer #1: Dear Authors,

I would like to congratulate the authors on this interesting study. Nevertheless, I would like to suggest some changes to the text.

Could the authors please provide sample size calculations and add them to the manuscript?

Was disability measured? This may have influenced the results.

Please explain why you chose 10 weeks of intervention? What was the reason?

Figure 1 is fuzzy - please improve it.

Could you translate all the references into English?

Reviewer #2: The present study aimed to investigate the impact of an intervention program focused on motor skills on the development of the motor, cognitive and social-emotional dimensions of infants and toddlers, within early care settings in Italy. The study is important and brings added value concerning the research of developmental issues at the early ages. Especially, the study on the children under 12 months could bring out insights on the possibility to intervene in order to contribute significantly to an integrated development (motor, cognitive and social-emotional).The study also investigates the impact of intervention if this is taking place indoor or outdoor. The results indicated that an outdoor intervention upon the motor skills development could contribute to the cognitive and social -emotional development for toddlers in a better way than an indoor intervention. This study has important strengths, including the relevance of the topic for early childhood education and care, the experimental design, the methodology used and the strategies for data analysis. Also, the authors presented some limits of the study and stated the possibility to overcome it by future research.

However, the manuscript has some limitations related to the introduction and discussion.

1. Introduction. The authors presented a good synthesis of the literature and the theories they based their research. Pointing out more studies done on the topic and emphasizing the differences between these studies will help for stronger arguments for the hypotheses.

2. Materials and methods

The methodology is detailed described and the authors mentioned the quality assurance of the intervention as well.

2.1 Participants. Information are extensively presented. How did you used the demographics data such as age and education of the parents?

2.2 Instruments. All the instruments are extensively presented and also their internal consistency.

2.3 Procedure. What are the main objectives of the intervention?

Since the I-MovE intervention is in the center of this study, the authors intended to demonstrate its efficiency and impact, therefore we expect to know more about the program, so, at least a short presentation of the targets motor skills/aims of each intervention session for the entire program could be added. Above all, this present study is about the effectiveness of this specific motor intervention program so it would be important for the practical reason to know more about the specific aims of each session.

3. Results. Tables and figures are significant and well presented. Also the results are descriptive and logically presented. Maybe it would be useful to follow the main hypotheses of the study.

4. Discussion. This part is structured and help the general understanding of the results. It would be useful to refer to more recent studies when discussing the results. The implications of the results could be discussed for different categories of target population (educators, psychologists, parents) and different settings.

5. Limits and conclusions. The authors analyse the limits and propose valid conclusions.

6. PLOS authors have the option to publish the peer review history of their article (what does this mean?). If published, this will include your full peer review and any attached files.

Reviewer #1: No

Reviewer #2: **Yes: **Aurora A Colomeischi

---

## [Author Response · Author response to Decision Letter 0]

15 Nov 2023

Dear Dr. Booth,

Thank you for your letter of September 1st regarding our manuscript, I-MovE (PONE-D-23-11988). An intervention to promote movement at childcare centers: benefits for motor cognitive and socio-emotional development”. We would like to thank you for giving us the opportunity to revise the paper. We are also grateful to the Reviewers for their comments, which have guided our revision and helped us improve the paper substantially.

In our response letter, we reiterate each of the issues you and the Reviewers raised (in italics) and detail how we have addressed each issue. Page numbers are provided. All revisions are written in red font in the marked-up copy to facilitate their identification.

We hope to meet with your approval. If further modifications are necessary, we are of course willing to make them.

Journal Requirements:

AUTHORS: The entire manuscript was revised according to the guidelines provided above.

"This research was funded by Joint Research 2021 grants promoted by the University of Verona." 

AUTHORS: We have modified the financial disclosure (see the cover letter) to acknowledge that the funder had no role in study design, data collection and analysis, decision to publish, or preparation of the manuscript. 

3.In your Data Availability statement, you have not specified where the minimal data set underlying the results described in your manuscript can be found. PLOS defines a study's minimal data set as the underlying data used to reach the conclusions drawn in the manuscript and any additional data required to replicate the reported study findings in their entirety. All PLOS journals require that the minimal data set be made fully available. For more information about our data policy, please see http://journals.plos.org/plosone/s/data-availability.

AUTHORS (ISSUES 3 AND 4): As required by the Journal and as explained in detail in the cover letter, we updated our Data Availability statement as follows: “The minimal anonymized data set supporting the conclusions of this article is available as a Supporting Information file”.

AUTHORS: We updated the reference list since we included more studies (i.e., 24, 31, 36, 37, 42-47, and 76 in the new reference list). We also reviewed the reference list and, as required by one of the reviewers, we translated the references which were not in English (i.e., references nr 61, 62, 70, 71). 

Reviewer #1

Dear Authors,

I would like to congratulate the authors on this interesting study. Nevertheless, I would like to suggest some changes to the text.

AUTHORS: Thank you for the time and effort in reviewing our paper and for recognizing the novelty of our work. We have tried to revise the manuscript based on your comments and suggestions to improve the paper’s quality. We carefully evaluated and responded to each of your comments/requests. We hope to meet with your approval. If further modifications are necessary, we are of course willing to make them.

Could the authors please provide sample size calculations and add them to the manuscript?

AUTHORS: In this revised version of the manuscript, we added sample size calculation analyses. In particular, we run post-hoc power analyses. First, for each LMM and for each Level, we estimated the η²p of the variables inserted in the model, with particular attention to the interaction term Group*Time which was crucial to test for the intervention’s effect. Within the “Results” section, we inserted the η²p for the main effects and interaction term in each of the LMM. 

Second and more importantly, in the Data Analysis section of the revised version of the manuscript, we also declared that “To test whether the sample sizes of the two groups (Level 1 and Level 2) in the present study play a role in limiting the significance of each Linear Mixed-Effects Model, a series of post-hoc power analyses were conducted. These analyses were performed using G*power [76], with an α = 0.05, an effect size (f) derived from the η²p of the interaction term Group*Time of each model, and the exact number of participants for Level 1 and Level 2.” (p. 14, l. 17-21). The power (1 − β) of the interaction term of each model for Level 1 and Level 2 analyses was specified in the Results section. For Level 1, we found an observed power (1 – β) of .99 for the Peabody Developmental Motor Scales; an observed power power (1 – β) of .30 for the EEF-RS, and a power (1 – β) of .75 for the Socio-Emotional Development Scale Caregiver Report. The power (1 – β) for Level 2 analyses was > .99 for all models (i.e., Peabody Developmental Motor Scales, EEF-RS, Brief-P, Socio-Emotional Development Scale Caregiver Report, Questionnaire for children’s social behaviour in nurseries,). Overall, the analyses revealed that the sample size in the present study did not play a role in limiting the significance of most of the models run. 

Considering the appropriate level of the post-hoc power analyses for almost all the models, the current version of the manuscript presents the results of the analyses first for Level 1 and then for Level 2 (please note that to address a request of the second reviewer - see comment 3 for further details - the results were also reorganized in other respects). In other words, we did not present analyses for Level 1 as explorative. However, we better specified and supported the discussion of the results on the effects of the I-MovE intervention on the cognitive and socio-emotional domain at Level 1 by considering the post-hoc power analysis of the EEF-Rating Scales which did not reach an appropriate power (p. 19, l. 29-31). This issue was also considered while discussing the limits of the study (p. 19; l. 35-38). Finally, we discussed the results for both Levels and all three domains by considering the effect sizes of the interaction terms and comparing them with those available in the literature (p. 18, l. 1, p. 19, l. 37-39). Overall, the effect sizes of the models, even after controlling for several intervening variables, were small to large and in line with the range reported in the Literature (see also p. 5 and 6). 

Was disability measured? This may have influenced the results.

AUTHORS: We agree with you that disability may have had an impact on the entire study and specifically on the results. However, based on teachers' and parents’ reports, none of the participants had been referred to the National Health Services for physical/motor disabilities and other disabilities or delays. In the revised version of the manuscript (p. 7, l. 47), we clarified that participants were typically developing children. 

Please explain why you chose 10 weeks of intervention? What was the reason?

AUTHORS: We chose a 10-weeks intervention based on various evidence. More generally, recent reviews and meta-analyses highlighted that there could be a wide variability among studies in the choice of the intervention duration, but interventions greater than 4 weeks exclude short-term effects (e.g., Van Capelle et al., 2017; Wick et al., 2017). More specifically, as highlighted by Riethmuller and colleagues (2009) in their review of the literature, the average duration of interventions to improve motor development in young children is 12 weeks. More recently, Hudson and colleagues (2021) pointed out that “18 studies reported dosage ranging from 5 to 60 hours” and that the median dosage reported is of 11.65 hours (this is in line also with Tortella et al., 2016 whose results are based on a 10-weeks intervention). In our study, teachers were required to implement the intervention twice a week for about 30-40 minutes (60-80 minutes per week) for 10 weeks. This results in a total of 600-800 minutes (corresponding to 10-13 hours with an average of 11.5 hours) of intervention, which is in line with previous studies. To better support our choice concerning the duration of the intervention, we added a short sentence at the end of the paragraph “Motor intervention program” (p. 10, l. 16-20): “The duration of the whole intervention program is greater than the duration recommended to exclude short-term effects and in line with the average duration of previously conducted motor programs (see, for example, Hudson and colleagues [6], Tortella et al. [47] for empirical evidence; see Riethmuller and colleagues [40], Wick et al. [42] and Van Capelle et al. [43] for reviews and a meta-analysis).”

Figure 1 is fuzzy - please improve it. 

AUTHORS: Thank you for pointing out this problem. An improved version of Figure 1 was provided in the revised version of the manuscript. 

Could you translate all the references into English? 

AUTHORS: We provided the English translation of all the references. In particular, we edited references number 50 (now 61), 51 (now 62), 59 (now 70) and 60 (now 71). 

Reviewer #2

The present study aimed to investigate the impact of an intervention program focused on motor skills on the development of the motor, cognitive and social-emotional dimensions of infants and toddlers, within early care settings in Italy. The study is important and brings added value concerning the research of developmental issues at the early ages. Especially, the study on the children under 12 months could bring out insights on the possibility to intervene in order to contribute significantly to an integrated development (motor, cognitive and social-emotional).The study also investigates the impact of intervention if this is taking place indoor or outdoor. The results indicated that an outdoor intervention upon the motor skills development could contribute to the cognitive and social -emotional development for toddlers in a better way than an indoor intervention. This study has important strengths, including the relevance of the topic for early childhood education and care, the experimental design, the methodology used and the strategies for data analysis. Also, the authors presented some limits of the study and stated the possibility to overcome it by future research.

However, the manuscript has some limitations related to the introduction and discussion.

AUTHORS: Thank you for the time and effort in reviewing our paper and for recognizing the strengths of our work. We revised the manuscript based on your comments and suggestions to improve the paper’s quality. We carefully evaluated and responded to each of your comments/requests. We hope to meet with your approval. If further modifications are necessary, we are of course willing to make them.

1. Introduction. The authors presented a good synthesis of the literature and the theories they based their research. Pointing out more studies done on the topic and emphasizing the differences between these studies will help for stronger arguments for the hypotheses.

AUTHORS: In the revised version of the manuscript, we pointed out more studies on the topic. In detail, we added a recent correlational study by Han and colleagues [31] (p. 4, from l. 14) to better support the relations between the motor, cognitive, and socio-emotional developmental domains. We also refer to the editorial published by Libertus and Haus (2017) [44] reporting on two studies investigating the effects of motor intervention on infants’ motor skills (see p. 5, from line 33). Moreover, we included a summary of the results of the review by Wiek et al., [42] and from the meta-analysis by Van Capelle and colleagues [43], analyzing the effects of gross-motor intervention on fundamental movement skills (p. 5, from l. 38). We highlighted that, although the studies included in these works considered children between 0 and 6 years, most of the studies promoted intervention in children between 3-5 years (see also other empirical studies, for example, Tortella et al., 2016). This further confirms the lack of intervention studies in infants and toddlers. Finally, we emphasized the characteristics of each study (e.g., the child’s age, the considered variables, the setting, the duration of the intervention, and the number of sessions) and which category of motor skills were considered in those studies and in the present one (see the Introduction and Aims and hypotheses sections; e.g., gross motor skills, locomotor skills). At this stage, we were not aware of other recent studies investigating the role of motor intervention on multiple developmental domains in young children. We hope our integrations meet the reviewer’s expectations, but we will consider any additional suggestion coming from the Reviewer. At the end of the letter, we provided a list of the new references.

2. Materials and methods

The methodology is detailed described and the authors mentioned the quality assurance of the intervention as well.

AUTHORS: Thank you for recognizing these methodological strengths.

2.1 Participants. Information are extensively presented. How did you used the demographics data such as age and education of the parents?

AUTHORS: The demographic data such as the parents’ age and level of education were considered together with other variables as potential intervening variables (see Table 1). These controls were necessary because of the design of the study (i.e., participating children were not randomly assigned to the intervention or control groups because of the difficulties of some nursery schools in adhering to and implementing the intervention; p. 8., l. 31-33. However, as reported in Table 1, for the parents’ age and their level of education we did not find any statistically significant differences between children in the experimental and control groups. We stressed this information more in the revised version of the manuscript (see p. 8, l. 44-49). We also added the statistic for the SVANI scale which was missing from the old version of the manuscript: “Based on these data, we can exclude differences among structures involved in recruiting experimental (IN or OUT) and control groups in terms of the quality of schools [F(2,15) =.583, p = .570, η²p = .07],” (p. 8, l. 45-46). Finally, for Level 2 only, there is a statistically significant difference among the three groups in terms of “hours of attending childcare”. We highlighted this finding on p. 8 line 46 of the manuscript, in the “Participants” section. For this reason, we re-run the analyses for Level 2 only, by considering “hours of attending childcare” as an additional covariate (together with “frequency of motor and outdoor activities, the restorativeness index of nursery schools”, as reported in the “Data Analysis” section, p. 14, l. 2-3). The results remain substantially unchanged with only one difference: the effect of the I-MovE intervention on the socio-emotional skills of children attending the program indoor became marginally significant (for more details, see answer to comment 3).

2.2 Instruments. All the instruments are extensively presented and also their internal consistency.

AUTHORS: Thank you for recognizing this methodological strength.

2.3 Procedure. What are the main objectives of the intervention?

Since the I-MovE intervention is in the center of this study, the authors intended to demonstrate its efficiency and impact, therefore we expect to know more about the program, so, at least a short presentation of the targets motor skills/aims of each intervention session for the entire program could be added. Above all, this present study is about the effectiveness of this specific motor intervention program so it would be important for the practical reason to know more about the specific aims of each session.

AUTHORS: Thank you for asking for this clarification. This request gave us the full chance to expand the description of the “Motor intervention program” section. As reported on p. 9 from line 36, we detailed the main aim and specific aims of the program. Moreover, we explained the competencies on which the program aimed to work each week. We also added that the details of the program are available upon request to the first author (see p. 10, l. 23-24). 

3. Results. Tables and figures are significant and well presented. Also the results are descriptive and logically presented. Maybe it would be useful to follow the main hypotheses of the study.

AUTHORS: In the current version of the manuscript, we refined the Data Analysis section, and we reorganized the Results section. After a further careful check of the manuscript, we have noticed that in the “Data analysis” section, the second research question has not been clearly stated. For this reason, we reformulated the sentence introducing how the results were presented “To test the effects of the I-MovE intervention program on experimental groups vs. control group (aim 1) and its effect in the group implementing the motor activities indoors vs the group implementing the activities outdoors (aim 2), two separate sets of analyses were run (first, for Level 1 and then, for Level 2) on the motor, cognitive, and socio-emotional skills” (p. 13, l. 37-41). In the data analysis section, we also added information regarding the consideration of the variable “hours of attending childcare” as a covariate in each of the run models for Level 2. 

To reorganize the result section, we followed the order of the two main aims of the study, and we presented the results concerning (1) the effect of the intervention and (2) effects based on the modality of implementation on each of the developmental domains (i.e., motor, cognitive, and socio-emotional development). Due to the results of the power analysis (see point 1 of Reviewer 1) showing that the sample size of Level 1 does not play a role in limiting the significance of the Linear Mixed-Effects Models in at least two out of three domains, analyses were run first for children in Level 1, and then for children in Level 2. Therefore, the results were reorganized into three main paragraphs addressing aims 1 and 2 for motor, cognitive, and socio-emotional skills, respectively. In the title of each of the three paragraphs, we specified the second hypothesis “and its modality of implementation (indoor vs. outdoor)” (e.g., “Effects of the I-MovE intervention program and its modality of implementation (indoor vs. outdoor) on children’s motor skills”; (e.g., p. 14, l. 32). Since we added a covariate to Level 2 models, we edited the statistics based on the new results. Table 3 (descriptive statistics) was also reorganized according to this structure. 

Finally, we modified the title paragraphs in the Discussion to better specify that the findings of both Aims and levels were discussed for each of the motor, cognitive, and socio-emotional domains (see p. 17, l. 38 and p. 18, l. 21).

Due to the numerosity of children in the experimental vs. control groups and the type of statistical test adopted, we cannot more explicitly differentiate the presentation of the results of the two specific aims since they are addressed in the same analysis. We hope these changes meet the reviewer’s requirements. We are open to following any other suggestion. 

4. Discussion. This part is structured and help the general understanding of the results. It would be useful to refer to more recent studies when discussing the results. The implications of the results could be discussed for different categories of target population (educators, psychologists, parents) and different settings.

AUTHORS: In this revised version of the paper, we discussed the main findings of the study by considering the literature reviewed in the Introduction (including the new references added in the revised version of the Introduction, as suggested in the first comment). In particular, we substantially extended the discussion on the effects of the I-MovE intervention on motor development (p. 17, from l. 41) and considered findings from more recent studies when discussing the effects of I-MovE intervention on cognitive and socio-emotional development (p. 18, from l. 25). Both reviews and metanalyses, and empirical works were considered to improve the Discussion. 

In the final part of the Limits and Conclusion section, we also added one paragraph to further elaborate on the practical implications of the results for psychologists, nursery school teachers, and parents. In particular, we elaborate on the application of the results in the clinical, school, and home settings (p. 20 from l. 36 and p. 21). 

5. Limits and conclusions. The authors analyse the limits and propose valid conclusions.

AUTHORS: Thank you for recognizing the analysis of the study’s limitations and the validity of our conclusions.

It follows the list of added references: 

11 Oakes, L. M., & Rakison, D. H. (2019). Developmental cascades. Building the infant minds. Oxford University Press.

24 Blair, C., & Raver, C. C. School readiness and self-regulation: A developmental psychobiological approach. Annual Review of Psychology 2015; 66, 711–731. https://doi.org/10.1146/annur ev-psych -01081 4-015221

31 Han, X., Zhao, M., Kong, Z., & Xie, J. Association between fundamental motor skills and executive function in preschool children: A cross-sectional study. Frontiers in psychology 2022; 13, 978994. https://doi.org/10.3389/fpsyg.2022.978994

36 Cheung, W.C., Shen, S. & Meadan, H. Correlation between Motor, Socio-Emotional Skills, and Academic Performance between Young Children with and without Disabilities. J Dev Phys Disabil 2022; 34, 211–231. https://doi.org/10.1007/s10882-021-09796-8

37 Bar-Haim, Y., & Bart, O. (2006). Motor Function and Social Participation in Kindergarten Children. Social Development, 15(2), 296–310. http://doi.org/10.1111/j.1467-9507.2006.00342.x

42 Wick, K., Leeger-Aschmann, C. S., Monn, N. D., Radtke, T., Ott, L. V., Rebholz, C. E., Cruz, S., Gerber, N., Schmutz, E. A., Puder, J. J., Munsch, S., Kakebeeke, T. H., Jenni, O. G., Granacher, U., & Kriemler, S. Interventions to promote fundamental movement skills in childcare and kindergarten: A systematic review and meta-analysis. Sports Medicine 2017; 47(10), 2045–2068. https://doi.org/10.1007/s40279-017-0723-1

43 Van Capelle, A., Broderick, C. R., van Doorn, N., E Ward, R., & Parmenter, B. J. Interventions to improve fundamental motor skills in pre-school aged children: A systematic review and meta-analysis. Journal of science and medicine in sport 2017; 20(7), 658–666. https://doi.org/10.1016/j.jsams.2016.11.008

44 Libertus, K., & Hauf, P. Editorial: Motor Skills and Their Foundational Role for Perceptual, Social, and Cognitive Development. Frontiers in psychology 2017; 8, 301. https://doi.org/10.3389/fpsyg.2017.00301

45 Wiesen, S. E., Watkins, R. M., & Needham, A. W. Active Motor Training Has Long-term Effects on Infants' Object Exploration. Frontiers in psychology 2016; 7, 599. https://doi.org/10.3389/fpsyg.2016.00599

46 Ryalls, B. O., Harbourne, R., Kelly-Vance, L., Wickstrom, J., Stergiou, N., & Kyvelidou, A. A Perceptual Motor Intervention Improves Play Behavior in Children with Moderate to Severe Cerebral Palsy. Frontiers in psychology 2016; 7, 643. https://doi.org/10.3389/fpsyg.2016.00643

47 Tortella, P., Haga, M., Loras, H., Sigmundsson, H., & Fumagalli, G. Motor Skill Development in Italian Pre-School Children Induced by Structured Activities in a Specific Playground. PloS one 2016, 11(7), e0160244. https://doi.org/10.1371/journal.pone.0160244

76 Faul, F., Erdfelder, E., Lang, A. G., & Buchner, A. G*Power 3: a flexible statistical power analysis program for the social, behavioral, and biomedical sciences. Behavior research methods 2007; 39(2), 175–191. https://doi.org/10.3758/bf03193146

---

## [Editor Report · Decision Letter 1]

10 Jan 2024

I-MovE. An intervention to promote movement at childcare centers: benefits for motor cognitive and socio-emotional development

PONE-D-23-11988R1

Dear Dr. Bastianello, 

We’re pleased to inform you that your manuscript has been judged scientifically suitable for publication and will be formally accepted for publication once it meets all outstanding technical requirements.

Kind regards,

Josephine N. Booth

Academic Editor

PLOS ONE
---

## [Editor Report · Acceptance letter]

19 Jan 2024

PONE-D-23-11988R1 

PLOS ONE

Dear Dr. Bastianello, 

I'm pleased to inform you that your manuscript has been deemed suitable for publication in PLOS ONE. Congratulations! Your manuscript is now being handed over to our production team.

Kind regards, 

on behalf of

Dr. Josephine N. Booth 

Academic Editor

PLOS ONE